# The Molecular Associations of Signet-Ring Cell Carcinoma in Colorectum: Meta-Analysis and System Review

**DOI:** 10.3390/medicina58070836

**Published:** 2022-06-21

**Authors:** Xueting Liu, Litao Huang, Menghan Liu, Zhu Wang

**Affiliations:** 1Department of Gastroenterology and Hepatology, West China Hospital, Sichuan University, Chengdu 610041, China; liuxueting89@163.com; 2Discipline Construction Department (Double-First Class Construction Office), West China Hospital, Sichuan University, Chengdu 610041, China; liumenghan@wchscu.cn; 3Department of Clinical Research Management, West China Hospital, Sichuan University, Chengdu 610041, China; huanglitao2423@163.com; 4The College of Literature and Journalism, Sichuan University, Chengdu 610041, China

**Keywords:** signet ring cell carcinoma, colorectal cancer, meta-analysis, systematic review

## Abstract

**Background:** Signet ring cell carcinoma (SC) accounts for 1% of total colorectal cancer (CRC) cases and is associated with aggressive behaviors, such as lymphatic invasion and distant metastases, resulting in poor prognosis. To date, there is still a lack of consensus on the genetic etiology underpinning this cancer subtype. This study aimed to clarify the molecular associations of SC by using meta-analysis and a systematic review. **Methods:** PubMed, Embase, and Cochrane Library were searched for studies evaluating the KRAS, BRAF, P53 statuses, and microsatellite instability (MSI) in CRC patients with different histological subtypes, including SC. The diagnosis of SC is defined as the signet ring cells comprising ≥50 percent of the tumor mass. By dividing the studies into subgroups based on the composition of control groups, such as classic adenocarcinoma (AC; no SC components) and non-SC (including those with SC components < 50%), the relative risk (RR) of molecular alterations for SC in each study were pooled using a random-effects model. Two reviewers identified trials for inclusion, assessed quality, and extracted data independently. **Results:** Data from 29 studies consisting of 9366 patients were included in this analysis. SC was associated positively with MSI (RR 1.78, 95% CI 1.34 to 2.37; 95% CI 0.77 to 4.15; *p* = 0.0005), BRAF mutation (RR 1.99, 95% CI 1.21 to 3.26; 95%CI 0.68 to 5.82; *p* = 0.0146), and negatively with KRAS mutation (RR 0.48, 95% CI 0.29 to 0.78; 95% CI 0.09 to 2.49; *p* = 0.0062). No association was found between SC and P53 expression (RR 0.92, 95% CI 0.76 to 1.13; 95%CI 0.61 to 1.39; *p* = 0.3790). Moreover, it was associated negatively with P53 gene mutations (RR 0.92, 95% CI 0.77 to 1.09; 95% CI 0.46 to 1.82; *p* = 0.1568), and P53 protein (RR 0.93, 95% CI 0.58 to 1.49; 95% CI 0.40 to 2.17; *p* = 0.6885). **Conclusions:** The molecular etiology of SC may be associated with the BRAF and MSI pathways. Its features, such as the high frequency of BRAF mutation, could partly explain its less favorable outcomes and limited effects of traditional chemotherapy.

## 1. Introduction

Colorectal cancer (CRC) is the third most common malignant tumor worldwide [1,2,3]. The histology of signet ring cell cancer (SC) is distinguished from classical adenocarcinoma (AC) by an abundant amount of intracellular mucin that displaces the nucleus [2]. While frequently diagnosed in the stomach, SC is a rare histological subtype of CRC, accounting for approximately 1% of cases [1,2,4,5]. The presence of SC in the colorectum is associated with young age, females, and more aggressive behaviors, including poorer differentiation, higher risk of lymphatic invasion, and distant metastasis [6,7,8]. As such, patients with SC were mostly diagnosed with advanced tumor stages and exhibited a significantly worse prognosis compared to those with AC, with the 5-year survival rate ranging from 12% to 20% [8]. This was further supported by the multivariable analysis implicating SC as an independent prognostic factor by including differentiation, TNM stage, lymphatic invasion, and angioinvasion as co-variates [8]. From a clinical perspective, careful assessment is undoubtedly needed for each CRC patient with SC.

The aggressive tumor biology of SC is assumed to be attributed to distinct genomic aberrations. KRAS, BRAF, the mismatch repair (MMR) gene, and microsatellite (MSI) status are the most common molecular markers routinely examined for therapeutic decision-making for CRC. While the presence of KRAS/BRAF mutations precludes the use of EGFR-targeted therapies, loss of MMR gene expression or MSI status is linked to the deficiency of the mismatch repair system and may inform the use of immunotherapy. Previous studies attempted to analyze the genetic etiology of different histological subtypes of CRC by evaluating molecular markers, including KRAS, BRAF, the mismatch repair (MMR) gene, and microsatellite (MSI) status. However, conflicting results were obtained, possibly due to the small sample size and the limited number of SC cases [9]. Although higher rates of MSI-H and BRAF mutation in SC were reported in several studies, no general agreement has yet been reached [2,4,6]. Moreover, tumors with less than 50% of signet ring cell components are not defined as SC according to the classification by the World Health Organization (WHO). However, it has been suggested that these tumors may share more similarities in molecular features with SCs [10], which may lead to bias among these comparative studies. For instance, BRAF mutation is reported as being closely associated with the presence of malignant SC, regardless of the percentages of signet ring cell components [11]. Given that recent data have suggested that treatment variance between SC and non-SC may lead to a disparity in survival with CRC, a thorough analysis of the molecular associations of SC in colorectum is needed.

Therefore, this study aimed to collect the available data on the molecular features of SC in the colorectum. Meta-analyses were performed to compare KRAS, BRAF, P53 status, and MMR status between SC and non-SC CRCs.

## 2. Methods

### 2.1. Literature Search and Study Selection

Our study was conducted in accordance with the PRISMA’s statements. The Addis, R, Endnote and other software used in the study complied with the relevant operational requirements. A systematic search of PubMed, Embase, and the Cochrane Library was performed, either for published or unpublished studies that compared KRAS, BRAF, P53 status, and MMR status between patients with SC and those with non-SC in the colorectum. We searched the literature by using the following search terms: (signet ring) AND (((colorectal) OR) colon) OR rectal and performed the latest search on 24 January 2022. The title and abstract of each citation were examined by two authors independently. Potential eligible studies were obtained in full text and assessed by each author. Disagreements were resolved by discussion, if needed, by a third author. Moreover, the reference lists of all included articles were further reviewed to find if there were any other eligible publications. Retrospective studies, single-arm studies, and those with no numerical data for the outcomes of interest, letters, comments, case studies, and editorials were excluded.

### 2.2. Eligibility Criteria

SC is defined by the presence of signet ring cells occupying more than 50% of the tumor area [7]. Comparative studies of SC and non-SC in colorectum with data of KRAS, BRAF, P53 status, and MSI/MMR were included. Studies that only included SC were excluded. Additionally, according to the classification of WHO, studies that did not define SC properly were excluded. Similarly, studies in which signet ring cell components were less than 50% were excluded. The primary outcome is KRAS status. Secondary outcomes include BRAF, P53, and DNA mismatch repair statuses. Last, during these procedures, no language restriction was applied.

### 2.3. Data Extraction and Management

Two reviewers (XT-L, LT-H) independently extracted data from studies that met the inclusion criteria using a pre-designed form, and disagreements were resolved by the third reviewer (ZW). The following data were extracted from each eligible study, including authors, year of publication, journal, countries in which the study was undertaken, the time during which the study was undertaken, the definition of SC diagnosis, the type of studies, numbers of patients with SC and non-SC of the colorectum, and KRAS, BRAF, MMR/MSI, and P53 status.

### 2.4. Statistical Analysis

A random-effect model, as described by Der Simonian and Laird [12], was used to calculate the pooled prevalence measures, risk ratios (RR) and 95% confidence intervals (CI). Heterogeneity between RRs for the same outcome was assessed by using both the Cochran’s Q test and the I^2^ statistic. Random-effects models of analyses were used if heterogeneity was detected (*p* < 0.10, I^2^ > 50%), otherwise, the fixed-effects model was used. A 2-sided *p* < 0.05 was considered to indicate statistical significance. Additionally, ninety-five percent confidence intervals (CI) were calculated.

All data were initially divided into two groups, the SC group (in which signet ring cells occupy more than 50% of the tumor area) and the control group. Previous studies implied that there were similarities in the molecular features between SC and tumors with signet ring cell components, however, did not exceed 50% [2,10]. The following subgroup analyses were planned, studies were divided into subgroups based on the composition of control groups as non-SC (including those with signet cell components < 50%) and classic AC (no SC components; hereafter referred to as the AC group). Funnel plots were used to assess the existence of publication bias, if possible. R (version 3.6.2) and meta package (version 4.9-9) were used in this meta-analysis. Two reviewers (XT-L, LT-H) independently assessed the methodological quality of each included study. The Newcastle–Ottawa Scale [13] (with 9 as the highest score) was used to examine the patient selection, comparability of the study groups, and assessment of exposure in all these studies. Any disagreements were resolved by discussion with the third author (ZW).

## 3. Results

### 3.1. Literature Review

A flow diagram of the study selection is shown in Figure 1. A total of 1787 studies were retrieved based on our searching strategy: 955 from PubMed, 799 from Embase, and 33 from Cochrane Library. After the removal of duplicates, the number was reduced to 1347. By title and abstract alone, 1227 articles were excluded. A total of 91 articles were considered ineligible for inclusion after reviewing the full text. Finally, a total of 29 articles consisting of 9366 patients were included in the analysis (Figure 1).

### 3.2. Study Characteristic and Quality Assessment

The remaining 29 articles with information on 9366 patients were finally included in this analysis. Among these studies, 13 investigated the KRAS mutation, 9 described the BRAF mutation, 17 researched the MSI-H mutation, and 8 investigated the P53 positive mutations. Details of the papers included in the review are available in Table 1. In addition, the Newcastle–Ottawa Scale [13] was used to evaluate these 28 articles and their scores were all above 5.

## 4. Outcome Measures

### 4.1. KRAS Status

Thirteen studies [2,6,10,14,15,16,17,18,19,20,21,22,23] comparing the KRAS mutation on 3487 patients (168 SC, 3319 non-SC) were eligible for the inclusion. SC was negatively associated with the KRAS mutation (RR 0.48, 95% CI 0.29 to 0.78; 95% CI 0.09 to 2.49; *p* = 0.0062) (Figure 2(1)). There was significant heterogeneity (I^2^ = 20%; *p* = 0.24). The subgroup analysis also showed a similar tendency. Five studies [2,6,10,15,17], which compared data on 2294 patients (74 patients with SC, 2220 patients with AC), were included. SC was associated negatively with the KRAS mutation when comparing to AC (RR 0.25, 95% CI 0.16 to 0.39) (Figure 2(2)). No significant heterogeneity was observed. (I^2^ = 0%; *p* = 0.93).

### 4.2. BRAF Status

Nine studies [2,6,10,11,15,17,22,24,25], including data for the BRAF mutations in 2713 patients (118 signet ring cell, 2595 non signet ring cell), were eligible. SC was associated positively with the BRAF mutations (RR 1.99, 95% CI 1.21 to 3.26; 95% CI 0.68 to 5.82; *p* = 0.0146) (Figure 3(1)). There was no significant heterogeneity (I^2^ = 0%; *p* = 0.54). Moreover, in the subgroup analysis of six studies [2,6,10,15,17,25], the BRAF mutations were compared in 2595 patients (76 patients with SC, 2519 patients with C-CRC). SC was associated positively with the BRAF mutations when comparing to AC (RR 2.41, 95%CI 1.14 to 5.08) (Figure 3(2)). There was no significant heterogeneity (I^2^ = 0%; *p* = 0.44).

### 4.3. P53 Status

Eight studies [2,6,10,14,19,26,27,28], with data on 1357 patients (89 signet ring cell, 1268 non signet ring cell) were included in the analysis for P53 status. SCs and were not associated with altered P53 expression (RR 0.92, 95% CI 0.76 to 1.13; 95%CI 0.61 to 1.39; *p* = 0.3790) (Figure 4). There was no significant heterogeneity (I^2^ = 0%; *p* = 0.88). The subgroup analysis of Tp53 gene expression, which included three studies [2,6,14] with 63 SCs and 611 C-CRCs, showed a similar outcome (RR 0.91, 95% CI 0.77 to 1.09) (Figure 4a). There was also no significant heterogeneity (I^2^ = 0%; *p* = 0.91). Moreover, the subgroup analysis of Tp53 protein expression, which included five studies [10,19,26,27,28] with 26 SCs and 657 C-CRCs, showed a similar outcome (RR 0.93, 95% CI 0.58 to 1.49) (Figure 4b). There was also no significant heterogeneity (I^2^ = 0%; *p* = 0.59).

### 4.4. DNA Mismatch Repair Status

The MMR status was determined by immunohistochemistry for MMR proteins or by a polymerase chain reaction-based assessment of the microsatellite status. Seventeen studies [2,10,14,17,19,22,25,29,30,31,32,33,34,35,36,37] that included data on 6412 patients (196 SCs, 6216 non-SCs) when comparing MMR status were eligible for inclusion. SCs were positively associated with dMMR (RR 1.78, 95% CI 1.34 to 2.37; 95% CI 0.77 to 4.15) (Figure 5(1)). There was no significant heterogeneity (I^2^ = 0%; *p* = 0.46). Furthermore, in the subgroup analysis, four studies [2,10,14,17] with data on 59 SCs and 2141 C-CRCs showed the same tendency (RR 2.24, 95% CI 1.92 to 2.63) (Figure 5(2)). There was no significant heterogeneity as well (I^2^ = 0%; *p* > 0.99).

## 5. Discussion

Although the histological type has not yet been considered as a determinant factor in the treatment of CRC to date, it became evident that the divergence among the CRC subtypes may lead to distinct survival outcomes. A main histological feature of several CRCs is the existence of mucin, which may remain within the cells (i.e., SC) or be secreted (i.e., mucinous carcinomas). While the influence of extracellular mucinous components on prognosis remains controversial, the association between SC and poor prognosis is rigid. The present study found that SC was associated positively with the BRAF mutation and dMMR/MSI-H, and negatively with the KRAS mutation compared to non-SC. Meanwhile, no significant differences in the p53 status among SCs and non-SCs was observed, which is consistent with a previous study [14]. These findings may hold clinical implications for the treatment of CRC.

CRC is thought to evolve through three main distinct mechanisms. First, chromosomal instability (accounting for 70–90% of CRCs), resulting from loss of heterozygosity at multiple tumor suppressor gene loci [38], develops a series of genomic events initiated by an APC mutation, followed by RAS activation or function loss of p53 [39]. The second mechanism is associated with RAS and RAF mutation, and also epigenetic instability (20–30%), resulting in abnormal methylation and the silencing of tumor suppressor genes, contributing to microsatellite stable and unstable cancers [39]. The third is MSI (2–7%). Mutations in MMR genes occurred in these tumors, which then resulted in an inability to repair the single-nucleotide DNA mismatches [38,39].

The findings from this meta-analysis support that SC is more likely to display the dMMR/MSI-H tumor genotype, which is consistent with its strong association with Lynch syndrome as previously described [8]. dMMR/MSI-H is generally considered a favorable prognostic factor associated with improved overall and disease-free survival in CRCs [15]. However, increased CRC-specific mortality is associated with the BRAF mutation, even in dMMR/MSI-H patients [40]. While a proficient MMR activity in CRC cells may restore the cytotoxic response to several chemotherapy drugs, such as 5-FU and platinum compounds, dMMR/MSI-H tumors implicated a clearly reduced response to traditional chemotherapy [41]. This may partially explain the poor response to treatment in patients with SCs. Recently, immunotherapy, such as PD-1 inhibitors, has been proved effective in patients with a dMMR/MSI-H status [42], providing a potential therapeutic strategy for SC in the colorectum.

Given the significant association of SC in the colorectum with distant metastases, the therapeutic decision-making for SC predominantly occurs in the setting of metastatic CRCs, wherein the anti-EGFR antibodies (e.g., cetuximab) remain to be the first-line choice. However, EGFR inhibitors may be less effective in the treatment of SCs due to the high prevalence of BRAF mutations. Previous studies have revealed that patients with BRAF-mutated tumors demonstrated a low level of response to anti-EGFR therapies [43]. Meanwhile, the KRAS mutation also plays a definitive role in the EGFR signaling pathway [23], which now has also been implicated as a determinant factor for the use of anti-EGFR antibodies [44,45]. In contrast to the positive correlation between mucinous carcinoma and the KRAS mutation, our results suggest that SC was associated negatively with the KRAS mutation, suggesting the distinct clonal origins of SC and mucinous carcinomas. Moreover, it has been suggested that the subgroup in which patients with BRAF-mutated and KRAS wild-type tumors also declines in later lines of therapy, such as EGFR inhibitors [43]. Collectively, the high frequency of BRAF mutations in SCs might partly explain the limited effects of EGFR inhibitors.

Several limitations should be noted. The significant heterogeneity in KRAS analyses implies that there may be methodological differences and publication bias among these studies. Furthermore, previous studies point out that tumors with even less than a 50% signet ring cell component are more similar to signet ring cell cancers in molecular features [10]. Since AC, with less than a 50% signet ring cell component, was defined as non-SC, according to the literature of WHO, this may neutralize the differences between the SC group and non-SC group. As such, we performed subgroup analyses and found a high frequency of BRAF mutations in SCs. In addition, MMR and MSI were considered equivalent in our studies. However, different methods were used to detect the MSI status and MMR, which might cause bias across these studies. Most of our included studies are retrospective studies and there may be selection bias and information bias.

Together, our study demonstrated that SC in the colorectum is associated positively with the BRAF mutation, dMMR/MSI-H status, and negatively with KRAS mutations. The molecular features and their intrinsic tumor biology of SCs may partly explain their limited response to current therapies and poor clinical prognosis. To date, there are still few studies looking into its molecular etiology, possibly due to its rarity. Further studies are warranted to explore its molecular mechanisms and potential therapeutic targets.

## Figures and Tables

**Figure 1 medicina-58-00836-f001:**
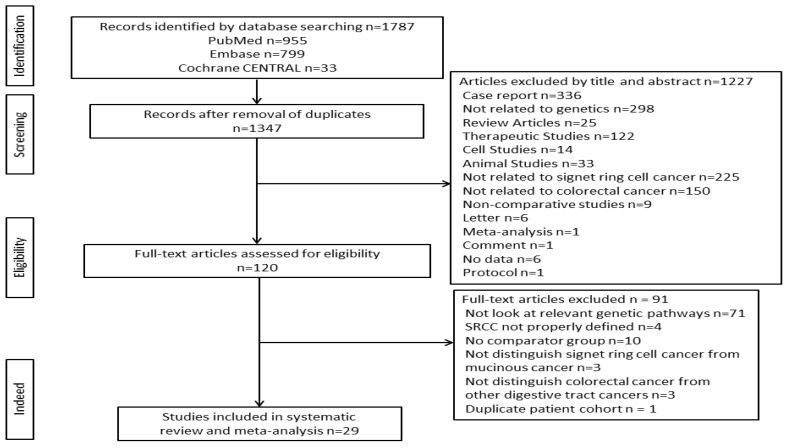
Flowchart for selection of eligible articles.

**Figure 2 medicina-58-00836-f002:**
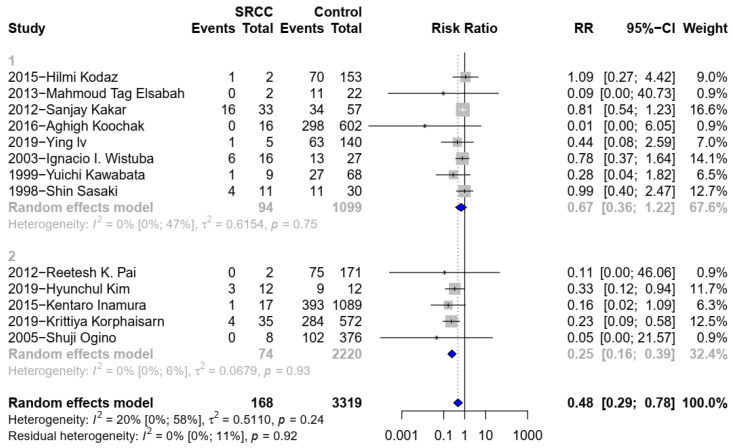
(**1**). Forest plot for KRAS between SRCC and non-SRCC (SRC component < 50% was not excluded). (**2**). Forest plot for KRAS between SRCC and C-CRC.

**Figure 3 medicina-58-00836-f003:**
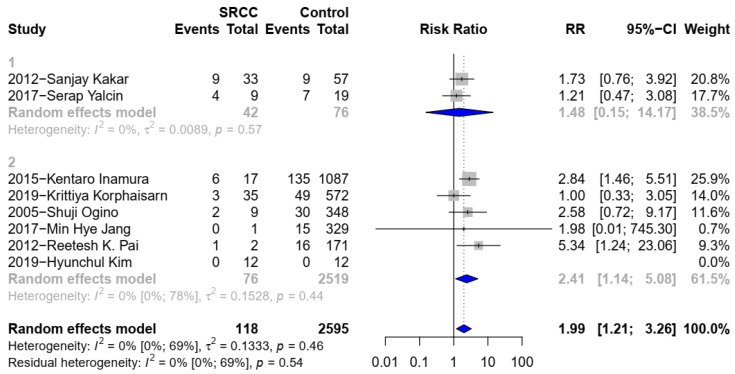
(**1**). Forest plot for BRAF between SRCC and non-SRCC (SRC component < 50% was not excluded). (**2**). Forest plot for BRAF between SRCC and C-CRC.

**Figure 4 medicina-58-00836-f004:**
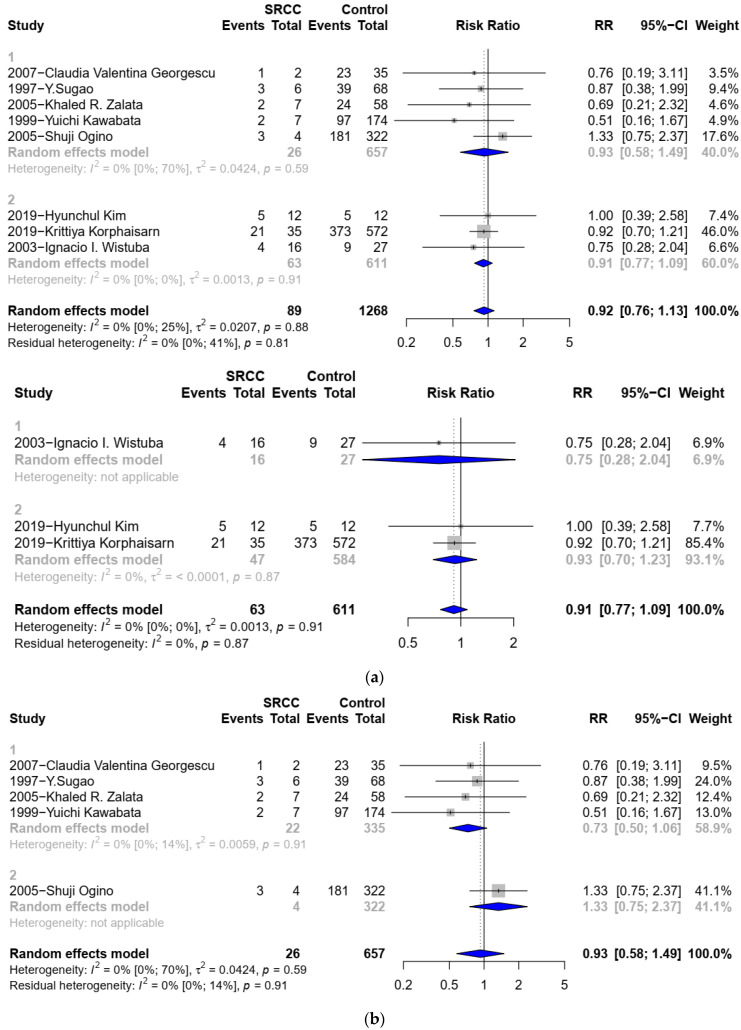
Forest plot for P53 status: (**1**). Forest plot for P53 status between SRCC and non-SRCC (using protein detection as test method). (**2**). Forest plot for P53 status between SRCC and non-SRCC (using gene detection as test method). (**a**) Forest plot for P53 status: (**1**). Forest plot for P53 status between SRCC and non-SRCC (SRC component < 50% was not excluded). (**2**). Forest plot for P53 status between SRCC and C-CRC. (**b**) Forest plot for P53 status: (**1**). Forest plot for P53 status between SRCC and non-SRCC (SRC component < 50% was not excluded). (**2**). Forest plot for p53 status between SRCC and C-CRC.

**Figure 5 medicina-58-00836-f005:**
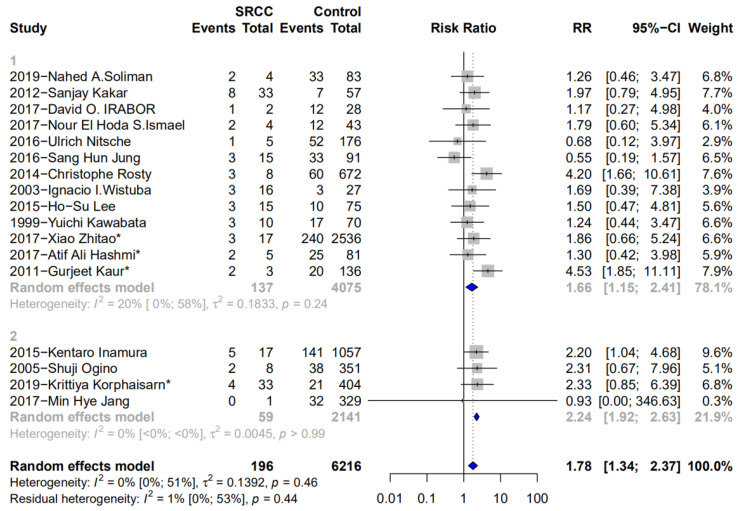
Forest plot for MSI-H status: (**1**) Forest plot for MSI-H status between SRCC and non-SRCC (SRC component < 50% was not excluded). (**2**) Forest plot for MSI-H status between SRCC and C-CRC (* represents these articles use MMR as test methods).

**Table 1 medicina-58-00836-t001:** Characteristics of included studies.

1st Author	Year	Country	Study Type	Enrolment Interval	Total	KRAS Mutated	BRAF Mutated	MSI-H	TP53 Positive	Ottawa Score
Hilmi Kodaz	2015	Turkey	Retro	2007~2014	189	90	N/A	N/A	N/A	6
Serap Yalcin	2017	Turkey	Retro	Not mention	28	N/A	11	N/A	N/A	7
Min Hye Jang	2017	South Korea	Retro	2011~2014	346	N/A	20	35	N/A	7
Reetesh K. Pai	2012	America	Retro	2005~2010	181	78	20	N/A	N/A	7
Shin Sasaki	1998	Japan	Retro	1964~1996	70	33	N/A	N/A.	N/A	8
Ho-Su Lee	2015	South Korea	Retro	2003~2011	90	N/A	N/A	13	N/A	8
Qing Wei	2016	China	Retro	2008~2015	61	N/A	2	N/A.	N/A	8
Xiao Zhitao	2017	China	Retro	2011~2015	2684	N/A	N/A	275	N/A	6
ClaudiaValentinaGeorgescu	2007	Romania	Retro	2005	41	N/A	N/A	N/A.	24	6
Ignacio I. Wistuba	2003	America	Retro	Not mention	43	19	N/A	6	13	7
Shuji Ogino	2005	America	Retro	Not mention	568	176	72	95	238	7
Hyunchul Kim	2019	South Korea	Retro	2003~2012	46	25	2	N/A	17	7
Mahmoud Tag Elsabah	2013	Egypt	Retro	Not mention	26	11	N/A	N/A	N/A	6
Nahed A.Soliman	2019	Egypt	Retro	2015~2018	115	N/A	N/A	54	N/A	6
Y.Sugao	1997	Japan	Retro	1963~1995	84	N/A	N/A	N/A	48	6
Sanjay Kakar	2012	America	Retro	Not mention	116	46	30	22	N/A	8
David O.IRABOR	2017	Nigeria	Retro	2007~2014	35	N/A	N/A	15	N/A	6
Atif Ali Hashmi	2017	Pakistan	Retro	2013~2015	100	N/A	N/A	34	N/A	6
Gurjeet Kaur	2011	Malaysia	Retro	2004~2007	150	N/A	N/A	28	N/A	6
Nour El Hoda S.Ismael	2017	Egypt	Retro	2012~2015	52	N/A	N/A	16	N/A	7
Yuichi Kawabata	1999	Japan	Retro	1981~1995	77	28	N/A	20	99	8
Aghigh Koochak	2016	Iran	Cross sectional	2008~2012	1000	336	N/A	N/A	N/A	Not Done
Khaled R. Zalata	2005	Egypt	Retro	2002~2004	75	N/A	N/A	N/A	29	6
Ulrich Nitsche	2016	Germany	Pros	1998~2012	256	N/A	N/A	89	N/A	8
Sang Hun Jung	2016	South Korea	Retro	2006~2012	176	N/A	N/A	56	N/A	7
KentaroInamura	2015	America	Pros	2008~2012	1220	440	179	190	N/A	7
ChristopheRosty	2014	Australia	Retro	1990~1994	738	N/A	N/A	86	N/A	8
KrittiyaKorphaisarn	2019	America	Retro	2009~2015	635	298	54	27	409	8
Ying lv	2019	China	Retro	2012~2017	164	72	N/A	N/A	N/A	6

## Data Availability

Not applicable.

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
