# Peer review of "The Molecular Associations of Signet-Ring Cell Carcinoma in Colorectum: Meta-Analysis and System Review"

_medicina, 2022, doi:10.3390/medicina58070836_

Round 1
Reviewer 1 Report
In present manuscript is entitled ‘The molecular associations of signet-ring cell carcinoma in colorectums: meta-analysis and system review’. The authors aimed to compare the four biomolecular features between SRC and non-CRC. In the meta-analysis study, the authors showed that SRC might be associated with the BRAF and MSI pathways. The manuscript needs some revisions.
My specific comments are appended as below:
1) According to existing literature (Run-Cong Nie et al.,), the SRC tends to affect young females. The authors may need to analyze the molecular associations with the gander based.
2) In the manuscript, the authors selected four biomarkers to study the SRC meta-analysis. However, these biomarkers are present in most the tumors, for example, KRAS mutations are present in ~25% of tumors (mostly lung cancer, and pancreatic cancer). The authors should write a paragraph in the introduction about why these KRAS, BRAF, p53, and MSI biomarkers are important in CRC.
3) The abstract is written well but the authors should maintain the unique scientific writing, for example, lane 18, BRAF gene mentioned without italic, and p53 written in different forms TP53, Tp53. The authors should check the abbreviations, for example, the CRC needs to be in lane 13 instead of lane 19.
4) Authors need to check all the figure descriptions. In the manuscript Fig2a $ 2b mentioned but in figure descriptions mentioned as Fig2.1 $ 2.2. Same with figure 3. Table 1 is missing in the manuscript.
5) The figures may need to be revised. For better understanding, the ‘control’ should be changed to non-SRCC or C-CRC.
Author Response
1)According to existing literature (Run-Cong Nie et al.,), the SRC tends to affect young females. The authors may need to analyze the molecular associations with the gander based.
Answer: We agree with the reviewer’s comment. The associations between SRC and young females have been addressed by several clinical studies [PMID: 23989057]. Unfortunately, after evaluation, we concluded that based on current available data, we could not perform gender-based subgroup analyses unless individual data were obtained for each study. In terms of the molecular associations with the gender, this issue have been comprehenively evaluated by a study based on TCGA cohorts [PMID: 27165743]. Following your kind suggestions, we have revised our description regarding the association between gender and SRC.
2) In the manuscript, the authors selected four biomarkers to study the SRC meta-analysis. However, these biomarkers are present in most the tumors, for example, KRAS mutations are present in ~25% of tumors (mostly lung cancer, and pancreatic cancer). The authors should write a paragraph in the introduction about why these KRAS, BRAF, p53, and MSI biomarkers are important in CRC.
Answer: Thanks for the reviewer’s comment.. Indeed, KRAS mutations are most commonly observed in pancreatic cancers (~73.55%), followed by CRC (~47.78%) and lung cancer (~31.55%). Meanwhile, KRAS, BRAF, mismatch repair (MMR) gene, and microsatellite (MSI) status are the most common molecular markers routinely examined for therapeutic decision-making for CRC. While the presence of KRAS/BRAF mutations preclude the use of EGFR-targeted therapies, loss of MMR gene expression or MSI status is linked to the deficiency of mismatch repair system and may inform the use of immunotherapy [PMID: 31631858]. Following your kind suggestions, we have revised our introduction accordingly.
3) The abstract is written well but the authors should maintain the unique scientific writing, for example, lane 18, BRAF gene mentioned without italic, and p53 written in different forms TP53, Tp53. The authors should check the abbreviations, for example, the CRC needs to be in lane 13 instead of lane 19.
Answer: Thanks for the reviewer’s comment. We have replaced writing with unique style,and CRC abbreviations has been added.
4) Authors need to check all the figure descriptions. In the manuscript Fig2a $ 2b mentioned but in figure descriptions mentioned as Fig2.1 $ 2.2. Same with figure 3. Table 1 is missing in the manuscript.
Answer: Thanks for the reviewer’s comment.. We have replaced Fig.2A with Fig.2-1, Fig.2B with Fig.2-2, and Fig3 with the same substitution
5) The figures may need to be revised. For better understanding, the ‘control’ should be changed to non-SRCC or C-CRC.
Answer: Thanks for the reviewer’s comment, but it is not ambiguous because there are differences between the control groups, which we explain in the comments below the picture
Reviewer 2 Report
This systematic review is aimed to explore the molecular associations of signet cell CRC (SC CRC) by using meta-analysis. The study is of clinical importance since SC CRC displays poor clinical outcomes. They found that the presence of BRAF mutation together with the MSI staus are the main factors that determine the prognosis of SC CRC.
The study is well designed and well presented. The results are clear and understandable. The figures all help the understanding of the results. They discuss their results in a moderate way.
However, one study (Allart M, Leroy F, Kim S, Sefrioui D, Nayeri M, Zaanan A, Rousseau B, Ben Abdelghani M, de la Fouchardière C, Cacheux W, Legros R, Louafi S, Tougeron D, Bouché O, Fares N, Roquin G, Bignon AL, Maillet M, Pozet A, Hautefeuille V; AGEO investigators. Metastatic colorectal carcinoma with signet-ring cells: Clinical, histological and molecular description from an Association des Gastro-Entérologues Oncologues (AGEO) French multicenter retrospective cohort. Dig Liver Dis. 2022 Mar;54(3):391-399. doi: 10.1016/j.dld.2021.06.031. Epub 2021 Aug 9. PMID: 34384712.) should be included into the analyses.
After minor corrections, I suggest accepting the manuscript for publication.
Author Response
Comments and Suggestions for Authors
This systematic review is aimed to explore the molecular associations of signet cell CRC (SC CRC) by using meta-analysis. The study is of clinical importance since SC CRC displays poor clinical outcomes. They found that the presence of BRAF mutation together with the MSI staus are the main factors that determine the prognosis of SC CRC.
The study is well designed and well presented. The results are clear and understandable. The figures all help the understanding of the results. They discuss their results in a moderate way.
However, one study (Allart M, Leroy F, Kim S, Sefrioui D, Nayeri M, Zaanan A, Rousseau B, Ben Abdelghani M, de la Fouchardière C, Cacheux W, Legros R, Louafi S, Tougeron D, Bouché O, Fares N, Roquin G, Bignon AL, Maillet M, Pozet A, Hautefeuille V; AGEO investigators. Metastatic colorectal carcinoma with signet-ring cells: Clinical, histological and molecular description from an Association des Gastro-Entérologues Oncologues (AGEO) French multicenter retrospective cohort. Dig Liver Dis. 2022 Mar;54(3):391-399. doi: 10.1016/j.dld.2021.06.031. Epub 2021 Aug 9. PMID: 34384712.) should be included into the analyses.
After minor corrections, I suggest accepting the manuscript for publication.
Answer: Thanks for the reviewer’s comment. Indeed, This article is relevant to our argument, but unfortunately we did not obtain individual information in this article and supporting documents, which made it difficult for us to conduct a combined analysis. However, allow us to discuss it in the discussion section.
Reviewer 3 Report
This peer-reviewed article presents the results and conclusions of a meta-analysis of the molecular associations of signet ring cell carcinoma (SC) in the colorectum. Based on the analysis of 29 strictly selected articles (a total of 9366 patients), the authors conclude that the molecular etiology of signet ring carcinoma involves the BRAF and MSI pathways. Despite some limitations mentioned by the authors, such as methodological differences and publication bias among the analyzed studies, still the described positive association of SC with BRAF mutation and dMMR/MSI-H status and negative with KRAS mutation should be taken into account by oncologists when planning patient examinations and proposed therapies. And this is why this meta-analysis should be published in the present form in the Medicina Journal.
The introduction is clear, concise, and informative. The materials and methods section describes the analysis performed very well, allowing other researchers to conduct such a meta-analysis. I have a small concern about repeating the data from the figures in the text of the results section, which would be inappropriate in the original research article, but in this case I would not have a simple suggestion on how to describe the data in a different way.
Minor comments:
Please explain the abbreviation used in the line 226 – is it 5-fluorouracil?
Author Response
This peer-reviewed article presents the results and conclusions of a meta-analysis of the molecular associations of signet ring cell carcinoma (SC) in the colorectum. Based on the analysis of 29 strictly selected articles (a total of 9366 patients), the authors conclude that the molecular etiology of signet ring carcinoma involves the BRAF and MSI pathways. Despite some limitations mentioned by the authors, such as methodological differences and publication bias among the analyzed studies, still the described positive association of SC with BRAF mutation and dMMR/MSI-H status and negative with KRAS mutation should be taken into account by oncologists when planning patient examinations and proposed therapies. And this is why this meta-analysis should be published in the present form in the Medicina Journal.
The introduction is clear, concise, and informative. The materials and methods section describes the analysis performed very well, allowing other researchers to conduct such a meta-analysis. I have a small concern about repeating the data from the figures in the text of the results section, which would be inappropriate in the original research article, but in this case I would not have a simple suggestion on how to describe the data in a different way.
Minor comments:
Please explain the abbreviation used in the line 226 – is it 5-fluorouracil?
Answer: Thanks for the reviewer’s comment.. Yes, it is 5-fluorouracil in the line 229。